# Optimization of the Amplification of Equine Muscle-Derived Mesenchymal Stromal Cells in a Hollow-Fiber Bioreactor

**DOI:** 10.3390/mps7020032

**Published:** 2024-04-02

**Authors:** Julien Duysens, Hélène Graide, Ariane Niesten, Ange Mouithys-Mickalad, Justine Ceusters, Didier Serteyn

**Affiliations:** 1Revatis SA, Rue de la Science 8, 6900 Marche-En-Famenne, Belgium; helene.graide@revatis.com (H.G.); justine.ceusters@revatis.com (J.C.); didier.serteyn@uliege.be (D.S.); 2Centre of Oxygen Research and Development (CORD), University of Liege, 4000 Liege, Belgium; ariane.niesten@uliege.be (A.N.); amouithys@uliege.be (A.M.-M.)

**Keywords:** cell expansion, bioreactor, mesenchymal stromal cells, horse, hollow fibers

## Abstract

The main causes of mortality in horses are the gastrointestinal pathologies associated with septic shock. Stem cells have shown, through systemic injection, a capacity to decrease inflammation and to regenerate injured tissue faster. Nevertheless, to achieve this rapid and total regeneration, systemic injections of 1 to 2 million cells per kilogram of body weight must be considered. Here, we demonstrate for the first time the feasibility and expansion capacity of equine muscle-derived mesenchymal stromal cells (mdMSCs) in a functionally closed, automated, perfusion-based, hollow-fiber bioreactor (HFBR) called the Quantum™ Cell Expansion System (Terumo Blood and Cell Technologies). This feature greatly increases the number of generated cells with a surface area of 1.7 m^2^. The expansion of mdMSCs is very efficient in this bioreactor. The maximum expansion generated twenty times more cells than the initial seeding in nine days. The best returns were observed with an optimal seeding between 10 and 25 million mdMSCs, using the Bull’s eye loading method and with a run duration between 7 and 10 days. Moreover, all the generated cells kept their stem properties: the ability to adhere to plastic and to differentiate into chondroblasts, osteoblasts and adipocytes. They also showed the expression of CD-44 and CD-90 markers, with a positive rate above 93%, while CD-45 and MHCII were non-expressed, with a positive rate below 0.5%. By capitalizing on the scalability, automation and 3D culture capabilities of the Quantum™, it is possible to generate large quantities of high-quality equine mdMSCs for gastrointestinal disorders and other clinical applications.

## 1. Introduction

Equine gastrointestinal disorders represent a significant challenge in equine medicine, with septic shock emerging as a critical complication. Strangulated obstructions are among the most severe conditions, necessitating emergency surgical intervention to restore intestinal positioning and organ vascularization. In the most extreme cases, enterectomy may be required [1,2,3]. These conditions can disrupt mucosal integrity, impair immune defenses and promote the translocation of bacteria and endotoxins across the intestinal barrier [3,4,5]. This can lead a septic shock, which is a life-threatening condition characterized by rapid systemic inflammation and organ dysfunction resulting from a dysregulated immune response [6]. While advancements in veterinary medicine have improved, equine practitioners continue to face diagnostic and therapeutic dilemmas due to the multifaceted nature of these conditions. Innovative approaches are needed to cure this pathology. In recent years, research into the therapeutic potential of mesenchymal stromal cells (MSCs) in equine medicine has expanded significantly. Mesenchymal stromal cells, derived from various sources, such as bone marrow, muscles, adipose tissue or umbilical cord blood, have demonstrated the ability to promote tissue repair, modulate inflammatory responses and enhance wound healing in preclinical and clinical studies [7,8,9]. These characteristics make them particularly attractive for the management of gastrointestinal disorders in horses [10]. Through systemic administration of a large quantity [11], such as 500 million, equine MSCs, it is possible to achieve widespread distribution of the cells throughout the body, allowing direct interaction with inflamed or injured gastrointestinal tissues, as well as the modulation of systemic immune responses. Using mdMSCs offers various benefits, such as easy access to muscle (which represents one-third of a horse’s body mass), less discomfort during sampling compared to aspirating bone marrow or adipose tissue and the rapid and reproducible nature of muscle microbiopsy across many animal species. Unfortunately, traditional amplification techniques such as T-flasks or spinner flasks are not sufficient to generate this number of cells. Our goal was to optimize the different parameters throughout the different trials to achieve fine-tuning and to generate a large number of mdMSCs using the Quantum™ Cell Expansion System. This bioreactor, composed of hollow fibers, presents numerous advantages over other bioreactor designs, like stirred tank or wave motion systems. First, we previously observed that mdMSCs require a physical support to achieve optimal amplification. Then, using the Quantum ensures that the mdMSCs are protected from the impeller and are further safeguarded inside the hollow fibers. The use of this bioreactor eliminates shear forces, leading to reduced cellular stress and improved viability. However, the attachment surface area remains limited, which constitutes a weakness compared to other bioreactor models, where the cells are in suspension. Finally, we decided to test the Quantum™ because this device has been successfully used to obtain fresh human mesenchymal stromal cells [12,13] but has never been used for mdMSCs. 

## 2. Material and Methods

### 2.1. Cell Cultures

#### 2.1.1. Equine Muscle Microbiopsy and Mesenchymal Stromal Cell Culture

The equine muscle-derived mesenchymal stromal cells used for the amplification were obtained via microbiopsy from the triceps brachii with local anesthesia, as described by Ceusters et al. [14], and stored at 4 °C for a maximum of 72 h. The study was conducted according to the guidelines and approved by the Animal Ethical Commission of the University of Liege (reference number 1609). The procedure on animals was approved by the Animal Ethical Commission of the University of Liège and was made following the relevant guidelines. Five horses were used as the donors for this amplification and differentiation study (one stallion, two geldings and two mares; their ages ranged between 8 and 22 years). Briefly, the 15 to 20 mg microbiopsies were washed with Phosphate Buffer Saline (PBS, Gibco, Grand Island, NY, USA), cut into small pieces and placed in a 24-well plate with DF-20 medium (DMEM/Ham’s F12 medium with 20% heat-inactivated FBS, 1% penicillin 1000 U/mL–streptomycin 10,000 µg/mL and 0.5% amphotericin B 250 µg/mL, all from Gibco, Grand Island, NY, USA) until confluence. The explant-derived cells were harvested using synthetic trypsin (TrypLE Express 1×, Gibco, Grand Island, NY, USA) and further isolated on a discontinuous Percoll (Gibco, Grand Island, NY, USA) gradient (3 layers: 15%, 25% and 35%). After centrifugation at 1250× *g* for 20 min without breaks, the cells between the 15% and 25% layers were collected, washed with PBS and resuspended in DF-20 culture medium. They were then further multiplied for up to 8 passages in T-175 cm^2^ flasks. The cells were freshly used or cryopreserved in a CryoStor^®^ CS5 solution with a concentration between 0.5 and 10 million (BioLife Solutions, Bothell, WA, USA) at −80 °C for short-term storage (up to 2 months) or in liquid nitrogen for the longer term (2 months to several years).

#### 2.1.2. Quantum™ Preparation and Amplification of the Equine Muscle-Derived Mesenchymal Stromal Cells

The Quantum™ Cell Expansion System (Quantum™ system; Terumo BCT, Inc., Lakewood, CO, USA) is a functionally closed hollow-fiber bioreactor (HFBR) system enclosed in a stand-alone incubator. It is designed for culturing both adherent and suspension cells. The surface area is 1.7 m^2^, which contains approximately 9200 hollow fibers (which is the equivalent of 100 T-flasks of 175 cm^2^) (Figure 1). The Quantum system fluid circuit is designed around two fluid loops for the intracapillary (IC) and extracapillary (EC) sides of the bioreactor. The bioreactor membrane allows free gas diffusion between the IC and EC sides of the bioreactor, as well as small molecule diffusion. Larger macromolecules are sequestered on the side of the membrane where they are added. The process, medium perfusion rate, harvest time, incubation time, and other tasks are controlled and adjusted by the user via a touchscreen surface.

##### Priming and Coating

The day before the seeding, a single-use cell expansion set (hollow fibers) was loaded onto the bioreactor device, which was pre-set at 37 °C, and the hollow fibers were coated overnight with a solution containing 5 mg of human fibronectin powder (Corning, New-York, NY, USA) diluted in 100 mL of PBS (50 µg/mL). The next day, 200 mL of PBS was used to wash the unbound fibronectin out of the system, and the PBS solution was totally replaced with DF-20 medium. A gas bottle which consisted of a mixture of 75% N_2_, 20% O_2_, and 5% CO_2_ was switched on just before the cell loading to obtain the optimal conditions for mesenchymal stem cell culture.

##### Loading and Feeding

Adherent mdMSCs were collected from the classical culture flasks as described above and transferred into the HFBR. Two methods of loading were tested. The classical method uses a high circulation speed during loading (200 mL/min). The alternative method, called “Bull’s eye”, begins with the same circulation rate and decreases that rate of circulation by half each round over the course of four recirculation events down to 25 mL/min, with each circulation event alternating in direction from the clockwise (positive) direction to the counterclockwise (negative) direction. Cell attachment periods occur between each circulation event to allow unattached cells the opportunity to bind to the coated membrane. The rocker was activated from 0° to 180° between each circulation event to allow a homogeneous cell distribution along the fiber surface and to reduce the presence of unwanted cells in the delivery tubes. The two loading methods were followed by a static attachment step. After this static period of adhesion (overnight), the cells were fed with fresh DF-20 medium at a feed rate of 0.1 mL/min, resulting in the passive removal of waste. The lactate concentration in the cell culture medium was measured every day (Accutrend^®^ Plus meter, Roche, Basel, Switzerland). The fresh medium rate was adjusted as a function of this lactate production rate according to the manufacturer’s instructions. A control T-flask 175 cm^2^ was seeded with the same proportional cell density as that in the bioreactor and incubated at 37 °C with 5% CO_2_. As an example, the flask was seeded with a total of 260,000 mdMSCs when the bioreactor was seeded with 25 million mdMSCs (both with a cell density of around 1470 cells/cm^2^).

##### Cell Harvest

A stationary lactate level indicates that the cell expansion phase is over. Approximately 24 h after reaching the final lactate threshold (between 1.2 and 1.6 mmol/L (data not shown)), the system was washed twice with 100 mL of PBS, and 200 mL of TrypLE Express was loaded into a cell inlet bag and connected to the expansion set. Four minutes of incubation time was sufficient to detach the adherent cells. PBS was used to flush the mdMSCs into a sterile harvest bag. This bag was sealed and placed into a laminar flow hood. The cells were collected from the harvest bag and centrifuged at 600× *g* for 5 min, the supernatant was discarded and the pellet was rinsed with PBS. After a second centrifugation step, the cells in the pellet were resuspended in CryoStor^®^ 5 (Merck, Saint-Louis, MO, USA) at a concentration of 10 million cells per mL for cryopreservation. A small volume of cell suspension was directly used for cell counting and cell viability.

### 2.2. Flow Cytometry Analysis of the Immunophenotype of the Generated Cells

Briefly, the mdMSCs were thawed, washed with PBS and centrifuged for 5 min at 600× *g*. The cell pellets were resuspended in 500 µL of FACS buffer (Miltenyi Biotec). A sample was taken to proceed to the cell counting and viability assessment. The cells were then incubated with conjugated antibodies coupled with their respective fluorochromes as follows and detailed in Table 1: CD44/*FITC* (Bio-Rad, Hercules, CA, USA), CD45/*PerCP* (Bio-Rad, Hercules, CA, USA), MHCII/*PE* (Bio-Rad, Hercules, CA, USA) or unconjugated (CD90) (Washington State University Monoclonal Antibody Center, Washington, WA, USA) primary antibodies. This was carried out for 15 min at 4 °C in the dark. The cells were diluted with FACS buffer and then centrifuged for 5 min at 600× *g*. Regarding the CD90 marker, a second antibody coupled with *FITC* (Abcam, Boston, MA, USA) was added to the primary unconjugated antibodies and incubated for 15 min at 4 °C in the dark. After 2 washes with FACS buffer, data were acquired using the MACSQuant Analyzer 10 (Miltenyi Biotech, Leiden, The Netherlands). 

### 2.3. Trilineage Differentiation of Generated Equine Mesenchymal Stromal Cells

The generated mdMSCs were seeded at a concentration of 200,000 cells/well in a 24-well plate filled with DF-20 medium. At confluence, DF-20 was, respectively, exchanged with 500 µL of StemPro^®^ Chondrogenesis, StemPro^®^ Osteogenesis and StemPro^®^ Adipogenesis Differentiation Medium for chondroblast, osteoblast and adipocyte differentiation. For the negative controls, the DF-20 medium was exchanged every 3–5 days. The plate was incubated for 14 days at 37 °C in a 5% CO_2_ incubator, and the differentiation medium was changed once a week according to the manufacturer’s instructions. Then, the differentiated cells were fixed with 4% formaldehyde and stained, respectively, with Alcian Blue, Alizarin Red and Oil Red O for 15 min to assess the presence of chondroblasts (mucopolysaccharides found in the cartilage matrix), osteoblasts (calcium deposits) and adipocytes (lipid content and especially triglycerides) (see Figure 2).

## 3. Results

### Muscle-Derived Mesenchymal Stromal Cell Culture, Amplification and Characterization

The mdMSCs of five different horses were successfully amplified using the Quantum™ device. A total of 10 to 25 million cells were seeded into the bioreactor at passages between 3 and 6. It appears that we obtained an optimal start with at least 10 to a maximum of 25 million fresh cells (Table 2). Indeed, frozen cells take two or three days longer to reach confluence. The best loading method is the ‘Bull’s eye” method; it allowed us to increase the amplification rate with an average multiplication factor of 11.19 (±5.50; n = 4), compared with 4.01 (±2.67; n = 4) for the classical loading method. Controlled culture T-175 cm^2^ flask seeding in parallel with the same proportional density as that in the bioreactor showed an average multiplication factor of 7.90 (±1.78; n = 8). The best expansion generated more than twenty times more cells than the initial seeding in nine days. The average population doubling time was over 4 days (4.26 ± 2.79 days). The lowest amplification was given in run 2 with a multiplication factor of 1.8, while run 1 showed a multiplication factor of 7.88, even though these runs originated from the same donor (horse 1). Run 2 presented the lowest cell viability with 60% viable cells. We also demonstrated that optimal run duration is between 7 and 9 days. 

The generated mdMSCs in the hollow-fiber bioreactor kept their characteristics (Figure 2), as recommended by the International Society for Cellular Therapy. They were able to undergo trilineage differentiation (adipogenic, osteogenic, chondrogenic) and were plastic-adherent (data not shown). They expressed the classical immunophenotype as shown in Figure 2. Amplified cells from runs 1, 4, 7 and 8 were analyzed using FACS and were positive for CD-44 and CD-90 markers with, respectively, an average positivity of 96.16 ± 1.77% and 97.29 ± 0.75%. CD-45 and MHCII markers had a very low expression at, respectively, 0.18 ± 0.11% and 0.13 ± 0.12%.

## 4. Discussion

We demonstrated for the first time that equine mdMSCs are able to adhere and expand in the Quantum™ Cell Expansion System. Eight runs were performed, and for each run, the number of mdMSCs was higher than the seeding. The best one generated more than twenty time more cells than the initial seeding, with a harvest of 220 million cells. The lowest amplification was observed for run 2 with a multiplication factor of 1.8. This can be explained by our decision to leave the cells for 15 days in the bioreactor to observe whether they continued to develop or not despite the lactate “plateau” phase. The maximal harvest generated 326 million cells, and this confirmed the hypothesis that the aim of 500 million cells is achievable, as described by Lambrecht et al. [12]. This team have yielded between 316 and 444 million human periosteum-derived stem cells with a seeding of 20 million stem cells using the Quantum™, while Russel et al. [13] have previously shown that human MSCs manufactured in a hollow-fiber bioreactor (HFBR) are physically and functionally equivalent in term of their properties to classical culture.

Various parameters were evaluated during these amplifications, and we were able to optimize them progressively to achieve the best amplification in the final trial. The optimal seeding is between 10 and 25 million cells at 37 °C using the Bull’s eye loading method with an amplification time ranging from 7 to 10 days.

All the amplified mdMSCs possessed the following characteristics, as recommended by the International Society for Cellular Therapy (ISCT) to be identified as a stem cells: a fibroblast-like shape, plastic-adherent under standard culture conditions, expression of a phenotypic panel and the ability to differentiate into osteoblasts, adipocytes and chondroblasts in vitro [15]. 

To reach the number of mdMSCs required for systemic injections, besides classic culture flasks, the use of a bioreactor seems to be an efficient alternative to achieve the desired performance. The amplification of the mdMSCs in bioreactors, such as the Quantum™, presents an innovative approach to large-scale cell production for regenerative medicine applications and gastrointestinal disorders. This closed system allows us to reduce the working time and limits the operator’s manipulations [11]. The Quantum™ offers a controlled and scalable platform for the culture and expansion of equine mdMSCs, providing a suitable microenvironment for maintaining cell viability, proliferation and functionality. By integrating advanced monitoring and automation technologies, this device allows for precise regulation of the culture parameters, such as temperature, oxygenation and nutrient supply (as a function of the lactate rate), to optimize cell growth and productivity. 

One of the key advantages of utilizing the Quantum™ bioreactor to amplify equine mdMSCs is its ability to support three-dimensional (3D) culture systems, which more closely mimic the native microenvironment of muscle tissue compared to traditional two-dimensional (2D) culture methods. Three-dimensional culture systems promote cell–cell interactions, extracellular matrix deposition and tissue-like organization, enhancing the regenerative potential and therapeutic efficacy of MSCs [16]. 

Furthermore, the scalability and automation capabilities of the Quantum™ bioreactor enable the cost-effective and time-efficient production of clinically relevant quantities of high-quality equine muscle-derived MSCs with reproducible characteristics, which is essential for the clinical translation and commercialization of cell-based therapies in equine medicine. This is particularly advantageous for meeting the demands of large animal clinical trials and the commercial manufacturing of cell-based therapies for equine patients. The ability to rapidly expand equine MSC populations while maintaining their stemness and potency is crucial to ensure the safety and efficacy of cell-based products for use in horses.

Nevertheless, the total duration of time required to produce this number of cells should not be underestimated. Indeed, 10 to 14 days are required to obtain a sufficient amount of cells before isolating the mdMSCs. Additionally, amplifications of the selected mdMSCs, first in T-flasks and after in a hollow-fiber bioreactor, can take 2 to 3 weeks, resulting in a duration time of 4 to 5 weeks to reach hundreds of millions cells.

In conclusion, the amplification of equine muscle-derived MSCs using the Quantum™ represents a promising strategy for advancing cell-based therapies for gastrointestinal disorders associated with septic shock in horses. Indeed, by capitalizing on the scalability, automation and 3D culture capabilities of the Quantum™ bioreactor, it is possible to generate large quantities of high-quality equine MSCs for clinical applications.

## Figures and Tables

**Figure 1 mps-07-00032-f001:**
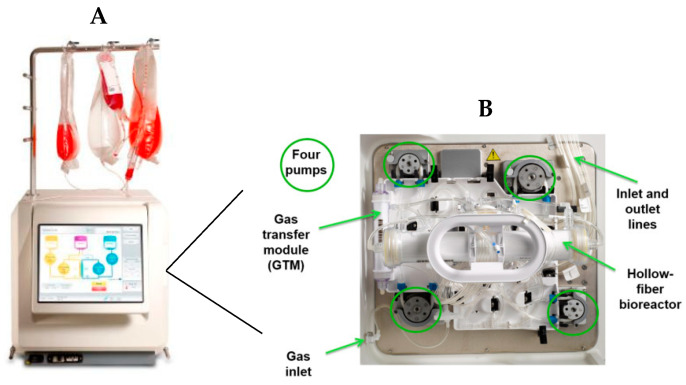
(**A**) The automated cell culture (Quantum™, Terumo BCT). (**B**) Internal view and description of the Quantum device loaded with the cell expansion set. This figure is a derivative of an image of the Quantum system (© Terumo BCT, Inc., Lakewood, CO, USA, 2013).

**Figure 2 mps-07-00032-f002:**
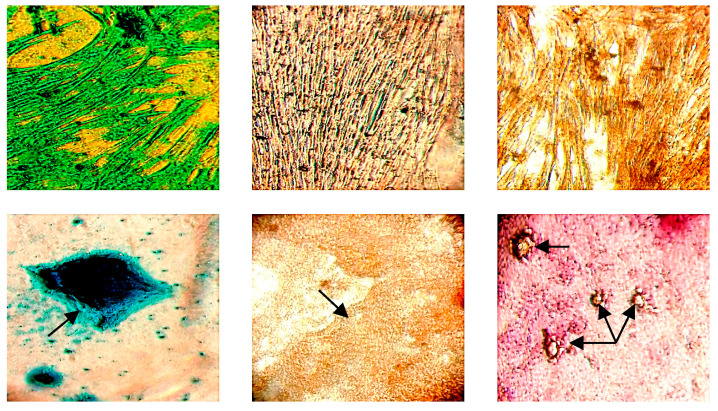
Representative microphotographs taken using a optical microscope (×100) of trilineage differentiations. Upper line is composed of (on the **left**) chondrocyte, adipocyte (in the **middle**) and osteocyte (on the **right**) cells controlled without differentiation media. Lower line is composed (on the **left**) of differentiated chondroblasts, adipocytes (in the **middle**) and osteoblasts (on the **right**).

**Table 1 mps-07-00032-t001:** Antibodies for analyzing the cellular proteins on mdMSCs.

Antibody	Clone	Dilution
CD-44	CVS18	25
CD-45	F-10-89-4	5
MHCII	CVS20	25
CD-90	DH24A	50

**Table 2 mps-07-00032-t002:** Cell amplification efficiency (8 runs) using the Quantum™ Cell Expansion System (Terumo BCT, Lakewood, CO, USA).

Run	Horse	Passage	Fresh/ Frozen	Loading	Seeding	Duration (Days)	Harvest	Cell Viability	Final Count (Viable Cells)	Multiplication Factor
1	Horse 1	P5	Fresh	Classic	25 × 10^6^	7	217 × 10^6^	91%	197 × 10^6^	7.88
2	Horse 1	P3	Fresh	Classic	10 × 10^6^	15	30 × 10^6^	60%	18 × 10^6^	1.80
3	Horse 1	P6	Frozen	Classic	16.5 × 10^6^	8	79 × 10^6^	73%	57.7 × 10^6^	3.50
4	Horse 2	P6	Fresh	Classic	25 × 10^6^	8	92 × 10^6^	78%	71.8 × 10^6^	2.87
5	Horse 3	P4	Fresh	Bull’s eye	25 × 10^6^	8	216 × 10^6^	83%	179 × 10^6^	7.16
6	Horse 3	P4	Frozen	Bull’s eye	20 × 10^6^	10	170 × 10^6^	82%	140 × 10^6^	7
7	Horse 4	P5	Frozen	Bull’s eye	25 × 10^6^	9	326 × 10^6^	91%	297 × 10^6^	11.88
8	Horse 5	P4	Frozen	Bull’s eye	10 × 10^6^	9	220 × 10^6^	85%	187 × 10^6^	18.70

## Data Availability

Data available on request due to restrictions, e.g., privacy or ethical. The data presented in this study are available on request from the corresponding author.

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
