# Peer review of "Optimization of the Amplification of Equine Muscle-Derived Mesenchymal Stromal Cells in a Hollow-Fiber Bioreactor"

_mps, 2024, doi:10.3390/mps7020032_

Round 1

Reviewer 1 Report

Comments and Suggestions for Authors

The manuscript by Duysens et al. describes an improved protocol for multiplying cell numbers of equine muscle-derived MSC using a hollow fiber bioreactor. Additionally, after multiple replications in the hollow fiber reactor, the authors demonstrate the potential of the manufactured cells to differentiate into chondroblasts, osteoblasts, and adipocytes, including the status of CD marker expression.

Despite the fact that the study is within the field of upscaling cellular products for cell therapeutic approaches, and therefore extremely important for the application in regenerative medicine, the study shows flaws regarding the M&M section.

The manuscript is clearly written and never loses its main thread.

First of all, the authors should consider renaming the cells. Please see the literature here: Ratajczak MZ, Zuba-Surma EK, Wojakowski W, Ratajczak J, Kucia M. Bone Marrow - Home of Versatile Stem Cells. Transfusion medicine and hemotherapy. 2008;35(3):248-259. doi:10.1159/000125585

It has been a while now that MSC are called mesenchymal stromal cells.

line 11: must be done? Please check – cells must be deployed?

lines 17 vs 113: 10 - 25 x10E6 cells were seeded; in the M&M section, 25 x 10E6, did the authors seed up to 25 x 10E6

line 67: Authors describe here the sample preparation; were the microbiopsies used for explant cultures of MSC?

lines 68-70: please check and add final concentrations DF-20 medium

line 95: fibronectin was used for coating; please could the authors add from which species the fibronectin was isolated?

lines 110-114: The cell density is described by the authors using precise numbers. Is this necessary? And if so, please check the number.

lines 123-124: is Cryostore a medium? Please add the company

lines 116-117: Please could the authors explain “after reaching the final lactate threshold”?

table 1: please could the authors comment on the column Dilution? 1:25?

lines 148-149: please add a brief protocol, which staining for which cells?

line 156: Table 2 does not show the maximum cell number authors mentioned in line 155; please check table number

line 165: please could the authors comment on the duration of run 2 and the outcome

figure 2: please include/provide images with higher resolution

lines 209-214: please could the authors comment on the 3D culture capabilities of a hollow fiber bioreactor compared to encapsulated cells as used in the cited paper

Please could the authors consider adding a paragraph about the duration time of cell manufacturing using the hollow fiber bioreactor? Biopsy samples, isolation of MSC, and subsequently expanding the cell mass. This could be crucial for the clinical translation.

Comments on the Quality of English Language

Please check for stem vs stromal cells 

Author Response

Please see in attachment

Reviewer 2 Report

Comments and Suggestions for Authors

This manuscript deals with amplification of equine muscle-derived mesenchymal stem cells in a hollow fibers bioreactor system. However, a lot of information is missing. Please see the comments below.

1.      The authors did not mention the reason to choose hollow fiber bioreactor to culture mdMSCs. The authors should compare the hollow fiber system with other bioreactors systems (e.g. stirred-tank, fixed-bed, wave motion, etc.). Advantages and disadvantages.

2.      In line 116, what is the value of the final lactate threshold? And what is the reason to harvest cells after the exponential phase? The cells may overgrow.

3.      The authors did not provide details of bioreactor cell culture control system, for example, the monitored temperature, pH, oxygenation, and nutrient supply in line 207. How well can this bioreactor system achieve?

4.      The authors mentioned the scalability of this system. The authors should provide the maximum surface area in this bioreactor system at the largest scale in commercial, compared to the current 1.7 m2, to estimate the maximum harvest cell number per batch. Also, the authors should prove the scalability of this bioreactor system is feasible.

5.      The authors use “Bull’s eye” loading methods to express the increased speed of fluids in fibers. Is it a trademark or registered name? Why not just use the high speed or low speed to distinguish them?

6.      The authors put optimization in the title. However, I did not see any optimization in the entire manuscript. For example, which factor is optimized and what is the optimal parameter should be used.

7.      In Table 2, Run#2 at Passage 3 showed the lowest viability 60%. The authors should give an explanation.

Author Response

Please see in attachment

Round 2

Reviewer 1 Report

Comments and Suggestions for Authors

Thank you very much for adressing the comments. Nevertheless, Figur 2 needs to be improved or another kind of presentation of the results. Please could authors consider to determine results via image analysis software like Image J. So, they could prepare a table to compare stainings of undifferentiated and differented cells. With this kind of table Fig 2 could be obsolet. Afterwards please check the text.

Comments on the Quality of English Language

please check spelling

Reviewer 2 Report

Comments and Suggestions for Authors

I don't have further questions. The revised manuscript is good for communication.

Author Response

/